

# Selenium uptake, tolerance and reduction in *Flammulina velutipes* supplied with selenite

Jipeng Wang[1,2,*], Bo Wang[3,*], Dan Zhang[1] and Yanhong Wu[1]

[1] Key Laboratory of Mountain Surface Process and Ecological Regulation, Institute of Mountain Hazards and Environment, Chinese Academy of Sciences, Chengdu, Sichuan, China
[2] University of Chinese Academy of Sciences, Beijing, China
[3] Soil and Fertilizer Research Institute, Sichuan Academy of Agricultural Sciences, Chengdu, Sichuan, China
* These authors contributed equally to this work.

## ABSTRACT

Recently, selenium (Se) enriched mushrooms have been exploited as dietary Se supplements, but our knowledge of the metabolic process during the Se enrichment process is far from complete. In this study, the uptake, tolerance and reduction of selenite in a widely cultivated mushroom, *Flammulina velutipes*, was investigated. The results showed that pH variation (from 5.5–7.5), metabolic inhibitor (0.1 mM 2,4-DNP) and P or S starvation led to 11–26% decreases in the selenite uptake rate of *F. velutipes*. This indicates that a minor portion of the selenite uptake was metabolism dependent, whereas a carrier-facilitated passive transport may be crucial. Growth inhibition of *F. velutipes* initiated at 0.1 mM selenite (11% decrease in the growth rate) and complete growth inhibition occurred at 3 mM selenite. A selenite concentration of 0.03–0.1 mM was recommended to maintain the balance between mycelium production and Se enrichment. *F. velutipes* was capable of reducing selenite to elemental Se [Se(0)] including Se(0) nanoparticles, possibly as a detoxification mechanism. This process depended on both selenite concentration and metabolism activity. Overall, the data obtained provided some basic information for the cultivation of the selenized *F. velutipes*, and highlighted the opportunity of using mushrooms for the production of Se(0) nanoparticles.

## INTRODUCTION

Selenium (Se) is an essential trace element for humans and animals. Supplementation of Se can be necessary because nutritional Se deficiency affects 500–1,000 million people worldwide, especially those from the Keshan disease area of China (*Combs, 2001*). The availability and biological activity of Se depend on its dose and chemical form (*Turło et al., 2011*). In trace amounts, Se confers antioxidant capacities to a number of selenoproteins (*Izquierdo, Casas & Herrero, 2010*). At higher concentrations, Se is toxic because it generates oxidative stress and is involved in DNA damage (*Izquierdo, Casas & Herrero, 2010*; *Máníková et al., 2010*). Organic Se-containing compounds may exhibit

Corresponding author
Yanhong Wu, yhwu@imde.ac.cn

anticarcinogenic activity, and their bioavailability to human beings and animals are considered to be high (*Chun et al., 2006*; *Suhajda et al., 2000*). Recently, there has been increasing interest in using selenized mushrooms as a source of Se supplementation (*Egressy-Molnár et al., 2016*; *Maseko et al., 2013*; *Rzymski et al., 2016*). Mushrooms have long been appreciated for their flavor and texture, as a nutritious food, and as a source of biologically active compounds (*Cheung, Cheung & Ooi, 2003*; *Yan & Chang, 2012*). Although the majority of mushrooms are Se-poor, a considerable amount of organic Se can accumulate in mushrooms supplied with selenite or other forms of Se (*Falandysz, 2008*). *Turło et al. (2011)* reported that the mycelium of *Lentinula edodes* cultured in selenite-fortified substrate accumulated Se in organic compounds, and the synergetic effect of Se compounds and active polysaccharides gave it a strong chemopreventive activity.

The transport of selenite through the plasma membrane is the first step of selenite metabolism (*Gharieb & Gadd, 2004*), whereas the mechanism of selenite uptake by mushrooms remains unclear. Selenite is present in the culture medium in different forms depending on the pH, including $H_2SeO_3$, $HSeO_3^-$ and $SeO_3^{2-}$, which are mediated by different transporters. In algae, both specific and nonspecific transport systems are involved in the selenite uptake process (*Araie et al., 2011*; *Morlon et al., 2006*; *Obata, Araie & Shiraiwa, 2004*). The specific one is driven by $\Delta$pH energized by $H^+$-ATPase (*Araie et al., 2011*), and both systems can be associated with transporters used by macronutrients such as phosphate, sulfate and nitrate (*Araie et al., 2011*; *Morlon et al., 2006*). In higher plants, selenite in the form of $H_2SeO_3$ is passively transported through aquaporins (*Zhang, Shi & Wang, 2006*; *Zhang et al., 2010*; *Zhao et al., 2010*), while $HSeO_3^-$ is absorbed in a carrier-mediated way, sharing common transporters with phosphate or sulfite (*Li, McGrath & Zhao, 2008*; *Zhang, Shi & Wang, 2006*). In yeasts, selenite is found to be absorbed in a metabolism dependent way, using the transporter of phosphate or monocarboxylate (*Gharieb & Gadd, 2004*; *Lazard et al., 2010*; *McDermott, Rosen & Liu, 2010*). Moreover, *Gharieb & Gadd (2004)* also reported a fast, metabolism-independent process during the selenite uptake of *Saccharomyces cerevisiae*. Mushrooms may share similar pathways of selenite uptake with these eukaryotic organisms, and thus medium pH, metabolism activity and competitive nutrients may serve as the major factors regulating the selenite uptake in mushrooms.

Mycelial growth and fructification of mushrooms can be hampered by toxic amounts of selenite in the growth substrate (*Dong et al., 2012*; *Niedzielski et al., 2015*; *Nunes et al., 2012*; *Turło et al., 2010b*). Therefore, an understanding of the tolerance of mushrooms to selenite is necessary to maintain the balance between mycelial production and Se enrichment. In addition to the growth changes when exposed to selenite, the mycelium of some mushrooms were reported to turn red due to the formation of Se(0) (*Gharieb, Wilkinson & Gadd, 1995*; *Turło, Gutkowska & Herold, 2010a*; *Vetchinkina et al., 2013*; *Vetchinkina et al., 2016*). Bioreduction of selenite to Se(0) has long been observed (*Levine, 1925*). Some bacteria and archaea grow anaerobically by linking the oxidation of organic substrates or $H_2$ to the dissimilatory reduction of Se oxyanions (*Lovley, 1993*; *Stolz & Oremland, 1999*). Other organisms, including bacteria (*Oremland et al., 2004*), fungi

(*Falcone & Nickerson, 1963*; *Gharieb, Wilkinson & Gadd, 1995*; *Nickerson & Falcone, 1963*) and plants (*Zhu et al., 2009*), can also reduce selenite to Se(0), most likely as a detoxification mechanism. The biologically formed Se(0) may partly exist as nanoparticles (*Vetchinkina et al., 2013*; *Vetchinkina et al., 2016*) which have novel biological activities (e.g., serve as an antioxidant, chemopreventive, and chemotherapeutic agent) and low toxicity (*Zhang, Li & Gao, 2012*). *Turło, Gutkowska & Herold (2010a)* suggested that Se(0), most likely in the nano-colloidal form, was responsible for the enhanced antioxidative activity of the mycelial extracts of the selenized *L. edodes*. Therefore, it is worthwhile to test mushrooms for the potentials of Se(0) production and to further examine the factors influencing this process.

*Flammulina velutipes* is among the four most widely cultivated mushrooms worldwide due to its desirable taste and nutritional values (*Jing et al., 2014*; *Yang et al., 2016*). With the high efficiency of Se accumulation (*Lin et al., 1997*), *F. velutipes* serves as a potential source of Se supplementation and biotransformation. In this study, we examined the uptake, tolerance and reduction of Se in *F. velutipes* supplied with 0–5 mM of selenite. In addition, the abilities of *F. velutipes* in selenite tolerance and reduction were compared with 11 other species of mushrooms. Thus, the relative sensitivity of *F. velutipes* to selenite and the universality of the selenite reduction process among mushroom species can be understood. The objectives of this study were: (1) to reveal the mechanisms of selenite uptake by *F. velutipes*; (2) to find out the optimal concentration of selenite for the cultivation of selenized *F. velutipes*; (3) to test the ability of *F. velutipes* to transform selenite to Se(0) and find out its influencing factors.

## MATERIALS AND METHODS

### Strains and culture conditions

The *F. velutipes* strain used in this study was obtained from the Soil and Fertilizer Institute, Sichuan Academy of Agricultural Sciences, China. Stock cultures were maintained on glucose-yeast (GY) agar plate consisting of glucose (20 g/L), yeast extract (5 g/L) and agar (18 g/L) at 4 °C in dark. Inoculum of 7 mM in diameter was picked up from the stock culture and inoculated onto the center of a GY agar plate (90 mM in diameter). After cultivated at 25 °C in the dark until 2/3 of the plate was covered by fungal colony, the marginal parts of the fungal colony were used as sources of inocula for selenite uptake, tolerance and reduction experiments. For the other 11 species of mushrooms (Tables S1 and S2), the same culture conditions were used as *F. velutipes*.

### Selenite uptake experiments

Mycelial pellets for the uptake experiments were obtained through shaking cultivation in the GY medium for 12 days (2 inocula of 7 mM in diameter). The composition of the nutrient solution used for selenite uptake (uptake solution) was 15 g/L glucose, 3 g/L arginine, 1.37 g/L KCl, 0.5 g/L MgCl$_2$ · 6H$_2$O. The pH of the solution was buffered at 6.0 with 2 mM MES (2-morpholinoethanesulphonic acid, pH adjusted with NaOH). After the uptake process, a desorption solution was used to get rid of the selenite adhering on the surface of the mycelial pellets. The composition of the desorption solution was

0.136 g/L $KH_2PO_4$, 0.172 g/L $CaSO_4 \cdot 2H_2O$. The solution was buffered at pH 6.0 with 2 mM MES and stored at 4 °C. To test the influence of pH on selenite uptake, the pH of the uptake solution was buffered at 5.5 and 6.5 with 2 mM MES, and at 7.5 with 2 mM HEPES (4-(2-hydroxyethyl)-1-piperazineethanesulfonic acid, pH adjusted with HCl). To test if the transporters of phosphate or sulfite were involved in selenite uptake, a P or S starvation treatment was conducted in a pretreatment solution before the uptake process. The composition of the pretreatment solution was 15 g/L glucose, 3 g/L arginine, 1.5 g/L $K_2HPO_4$ 0.6 g/L $MgSO_4 \cdot 7H_2O$ for the control treatment. The pH of the solution was buffered at 6.0 with 2 mM MES. In the −P or −S treatment, the $K_2HPO_4$ or $MgSO_4 \cdot 7H_2O$ were replaced by the corresponding chloride salts.

After shaking cultivation for 12 days, 30 mycelial pellets (5–10 mM in diameter) were gently picked up with tweezers, washed with 100 mL sterile water and 100 mL uptake solution, and then transferred into a 250 mL Erlenmeyer flask containing 100 mL uptake solution. After an adaption process of 30 min, 1 mL and 1 mM $Na_2SeO_3$ (sterilized with a 0.2 μm filter) was added to the Erlenmeyer flask to obtain a final selenite concentration of 0.01 mM and start the uptake process. The flask was then stoppered and shaken at 120 rpm and 25 °C for 60 min. The uptake process was stopped by transferring the mycelial pellets to the desorption solution (4 °C). After desorption for 15 min, the mycelial pellets were sopped up with filter paper and dried at 50 °C. For the 2,4-DNP (2,4-Dinitrophenol) treatment, 100 μL and 0.1 M 2,4-DNP dissolved in ethanol was added to the uptake solution (a final 2,4-DNP concentration of 0.1 mM) after the adaption process. Thirty minutes later, the uptake process was started as stated above. An additional control treatment of 0.1% (v/v) ethanol was included. For the −P or −S treatment, the mycelial pellets were cultivated in the pretreatment solution for 24 h before transferring into the uptake solution. A control treatment without P or S starvation was included. All the uptake experiments were performed under sterile conditions. Four replicates were conducted for each treatment.

The oven-dried mycelial pellets were grounded in an agate mortar. A subsample of ∼0.1 g was weighted with an analytical balance (± 0.0001 g) and digested with 8 mL 68–70% $HNO_3$ in a microwave oven (CEM Mars 6, CEM, Matthews, NC, USA). A standard reference material (Full name: CRM Citrus Leaf, GBW 10020; Se concentration: 0.17 ± 0.03 mg Se/kg; produced by: Institute of Geophysical and Geochemical Exploration, Chinese Academy of Geological Sciences, China) and blank samples were digested together with the mycelial pellets. The Se concentration in the solution was determined using an inductively coupled plasma-mass spectrometry (ICP-MS, NexION 300X; Perkin Elmer, Waltham, MA, USA).

## Selenite tolerance and reduction experiments

The cultivation of *F. velutipes* was carried out in the GY solid and liquid media supplemented with 0–5 mM $Na_2SeO_3$. Selenite was added to the culture medium at 50–55 °C (solid medium) or room temperature (liquid medium) from a stock solution (1 M) after sterilized with a 0.2 μm filter.

The tolerance experiment was conducted in solid and liquid media with initial selenite concentrations of 0–5 mM. The 9 selenite concentrations tested were

0, 0.001, 0.01, 0.03, 0.1, 0.3, 1, 3 and 5 mM. For solid cultivation, an isolate (7 mM in diameter) was inoculated onto the center of test plate containing 0–5 mM selenite, and incubated at 25 °C in the dark. The static cultivation was conducted in a 50 mL flask containing 20 mL of medium and was inoculated and cultivated in the same way as the solid cultivation. The biomass of the 20-day-old mycelia was determined after oven-dried at 60 °C. The shaking cultivation was conducted in a 250 mL flask containing 100 mL of medium with 2 inocula. After cultivation at 25 °C and 120 rpm in the dark for 13 days, the biomass of the mycelial pellets was determined after oven-dried at 60 °C. The radical growth rate, density, height, pigment secretion and other colony characteristics were recorded daily. Each treatment was performed with four replicates for the shaking cultivation and with five replicates for the solid and static cultivations. For the other 11 species of mushrooms, the tolerance experiments were conducted in solid cultivation with a selenite concentration of 0.1 mM in the same way as *F. velutipes*.

For the reduction experiment, the ability of *F. velutipes* to reduce selenite to Se(0) was determined visually, and the degree of red coloration resulted from Se(0) formation was used as an indication of reduction (*Gharieb, Wilkinson & Gadd, 1995*; no coloration, pink, pale red, and red represented no, weak, moderate, and strong reduction, respectively). During the tolerance experiment with initial selenite concentrations of 0–5 mM, the degree of red coloration of the fungal colonies was recorded daily. In addition, the reduction ability was examined after the full development of the mycelium. For this purpose, *F. velutipes* was first cultivated in selenite-free medium for 14 days until selenite was added to the medium to a final selenite concentration of 0–3 mM. Each treatment was performed in triplicate. The coloration of the mycelial pellets was observed hourly for the first 10 h and again at 24 and 32 h. All of the tolerance and reduction experiments were conducted under sterile conditions. For the other 11 species of mushrooms, the selenite reduction was determined visually in solid cultivation (0.1 mM selenite) and shaking cultivation (0.3 mM selenite).

For the TEM (transmission electron microscope) studies, the mycelia were picked from the red region of a 20-day-old colony (static cultivation, 0.3 mM selenite) with an inoculating needle and washed 3 times with a 0.01 M PBS solution (NaCl 8.01 g/L, KCl 0.20 g/L, $Na_2HPO_4 \cdot 12H_2O$ 3.58 g/L, $KH_2PO_4$ 0.27 g/L; pH 7.4). The mycelia were then suspended in a 1.5 mL centrifuge tube containing 0.5 mL of water and scattered in an ultrasonic cleaner. A drop of the obtained mixture was placed onto the copper grids using a rubber head dropper. After drying the sample, a transmission electron microscope (FEI Tecnai G2F20 S-TWIN; FEI, Hillsboro, OR, USA) working at 200 kV was used for morphology and energy dispersive X-ray spectroscopy (EDX) analysis.

### Statistical analysis

Comparisons among the mycelial biomass in static and shaking cultivations were performed by analysis of variance (ANOVA) followed by Duncan's multiple range test. The mycelial growth after the initial adaption period in the solid cultivation was fitted by a linear regression, and the differences between the slope ($k$) of the control treatment and

the selenite treatment were tested (Table S3). The SAS 9.1 (SAS Institute, Inc., Cary, North Carolina, USA) software package was used for all of the statistical analysis.

## RESULTS

### Influences of pH, metabolic inhibitor and nutrient starvation on the selenite uptake

The rate of selenite uptake was not significantly influenced by the medium pH ($p > 0.05$), yet its mean values decreased by 14% as the medium pH increased from 5.5–7.5 (Fig. 1A). The addition of 0.1 mM 2,4-DNP significantly inhibited the selenite uptake by 11% ($p < 0.05$) (Fig. 1B). The rate of selenite uptake responded similarly to the P and S starvation, and decreased by 25 and 26% after the P and S starvation, respectively ($p < 0.05$) (Fig. 1C).

### Growth responses of *F. velutipes* to 0–5 mM selenite

The growth rate of *F. velutipes* in the solid cultivation began to be inhibited at 0.1 mM selenite (growth rate decreased by 11%) (Fig. 2A; Table S3). The inhibition increased with the increasing selenite concentration. The mycelial growth stopped after 5 days in the media containing 1 mM selenite (Fig. 2A), and the colony margins became corral-like (Fig. S1). No mycelial growth was observed at the selenite concentrations of 3 mM or higher (Fig. 2A). After different durations of exposures to 3 or 5 mM selenite, the inocula were then re-inoculated to the selenite-free media. Longer lag periods were observed for the selenite stressed inocula, whereas the growth rates after germination were not severely impaired, except when the inocula were exposed to 5 mM selenite for 65 days (Fig. 2B; Table S4). In the static cultivation, the inhibition of mycelial growth started when the selenite concentration reached 0.1 mM, and the inhibition intensified as the selenite concentration increased. No mycelial growth was observed in the media containing selenite concentrations of 3 mM or higher (Fig. 3C). In the shaking cultivation, the responses of mycelial growth were similar to those in the static cultivation, except that significant growth inhibition started at the selenite concentration of 0.3 mM (Fig. 3D).

### Reduction of selenite to Se(0) by *F. velutipes* exposed to 0–5 mM of selenite

When *F. velutipes* was cultivated in the media containing 0–5 mM selenite (Figs. 3A–3C), the reduction intensities, as indicated by the red coloration, increased with increasing selenite concentrations in the range of 0–0.3 mM. Faint reduction appeared at the selenite concentration of 0.03 mM and the reduction intensity increased until 0.3 mM. The reduction intensity decreased at the selenite concentration of 1 mM. At the selenite concentrations of 3 mM or higher, there was no sign of mycelial growth or selenite reduction. When the mycelial pellets cultivated in the selenite-free media for 14 days were subjected to 0–3 mM selenite, the reduction intensity increased with increasing selenite concentrations (Fig. 3D). In the solid and static cultivation, denser colors were observed in the central part of the colony (around the inocula) compared to the margins (Figs. 3A, 3B and S2). In the shaking cultivation, denser colors were observed in the large

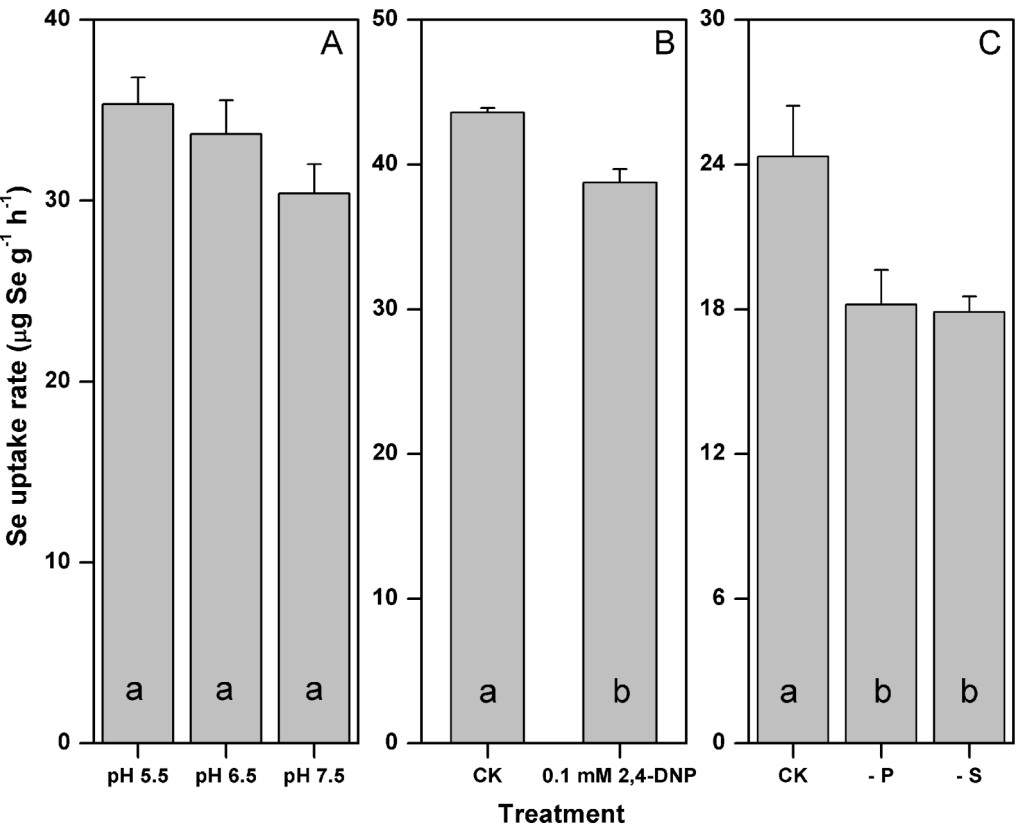

**Figure 1 Influences of pH (A) metabolic inhibitor (B) and P or S starvation (C) on the selenite uptake by *F. velutipes*.** Different letters indicate statistical difference at the 0.05 level.

mycelial pellets, especially those developed from the inocula, than in the newly formed small pellets (Figs. 3C, 3D and S2).

## DISCUSSION

### Possible mechanisms of selenite uptake by *F. velutipes*

The changes in selenite forms induced by the varying pH did not significantly influence the selenite uptake by *F. velutipes* (Fig. 1A). From pH 5.5–7.5, the proportion of $HSeO_3^-$ decreased markedly from 92.6–11.2%, corresponding to the increase of $SeO_3^{2-}$ from 7.3–88.8% (Table S5; the proportions of different forms were calculated based on the $pK_a$ values of selenous acid which determine the degree of the protonation of selenite under different pH values: $pK_1 = 2.57$, $pK_2 = 6.60$), whereas the selenite uptake rate only decreased by 14%. The lack of sensitivity of selenite uptake to medium pH of this range was commonly reported in organisms including algae (*Araie et al., 2011*; *Morlon et al., 2006*) and higher plants (*Zhang, Shi & Wang, 2006*; *Zhang et al., 2010*; *Zhao et al., 2010*). Thus, when the pH of medium is higher than 5, selenite speciation may not be the key regulator of selenite uptake. However, it was evident that in our and other studies (*Araie et al., 2011*; *McDermott, Rosen & Liu, 2010*; *Zhang, Shi & Wang, 2006*; *Zhang et al., 2010*; *Zhao et al., 2010*), selenite uptake showed a decreasing trend as medium

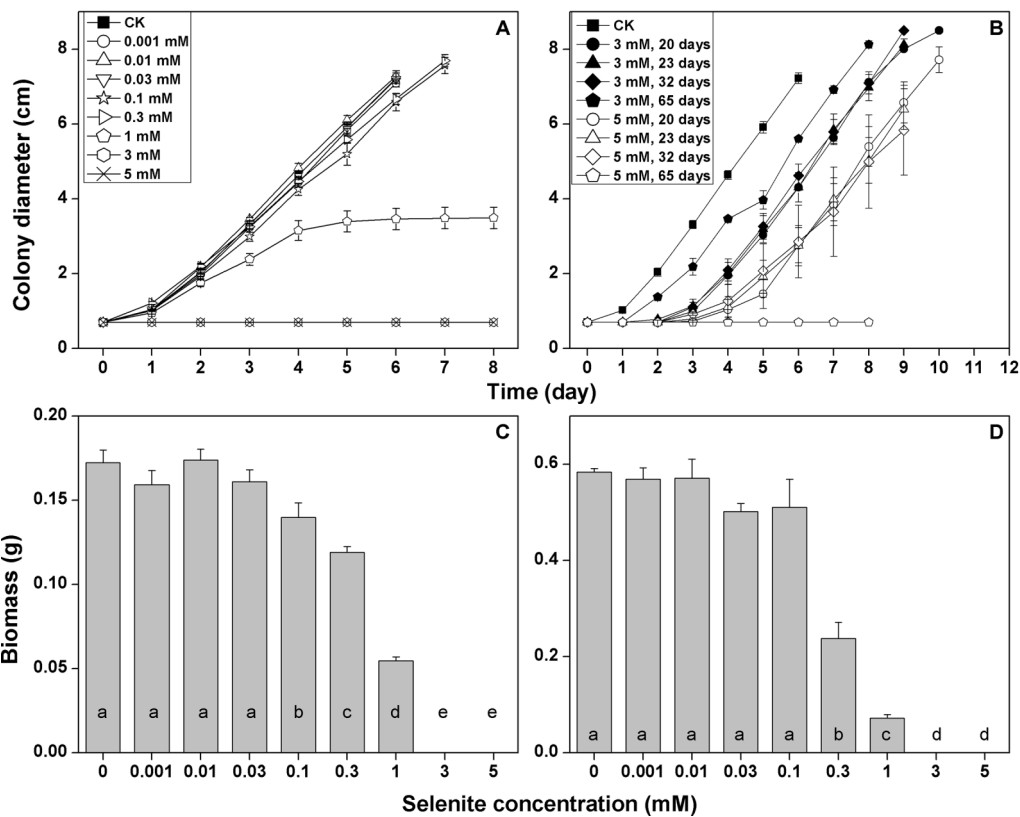

**Figure 2 Growth responses of *F. velutipes* to 0–5 mM selenite in solid cultivation (A and B) static cultivation (C) and shaking cultivation (D).** (D) the inocula were exposed to 3 or 5 mM selenite in solid cultivation for 20–65 days and were then inoculated onto the selenite-free media. Different letters indicate statistical difference at the 0.05 level.

pH increased from 5–8. This may originate from the selenite-proton symport according to the findings of *McDermott, Rosen & Liu (2010)*. Therefore, in *F. velutipes*, a small portion of the selenite might be taken up through a proton-coupled manner.

A minor fraction of the selenite was absorbed metabolically by *F. velutipes*. During our short-term uptake experiment, 2,4-DNP (the uncoupler of oxidative phosphorylation) inhibited the selenite uptake by 11% (Fig. 1B). In algae and yeast, the ΔpH generated by $H^+$-ATPase has been considered as the driving force of the active selenite transport (*Araie et al., 2011*; *McDermott, Rosen & Liu, 2010*). In combination with the responses of selenite uptake to medium pH, the active selenite absorption by *F. velutipes* may be associated with the anion-proton symporters. The inhibition rate observed in this study was lower compared with similar studies using 2,4-DNP or CCCP (carbonyl cyanide m-chlorophenyl hydrazone) as metabolic inhibitors at near neutral pH (inhibition rates ranging from ~25–80%) (*Gharieb & Gadd, 2004*; *Li, McGrath & Zhao, 2008*; *Zhang et al., 2010*). This highlighted the significance of the passive transport in the selenite uptake by *F. velutipes*. *Gharieb & Gadd (2004)* observed a fast, metabolism-independent phase in the selenite uptake process of *S. cerevisiae*, and this phase was responsible for the majority of the absorbed Se. The authors attributed this phase to abiotic adsorption and simple

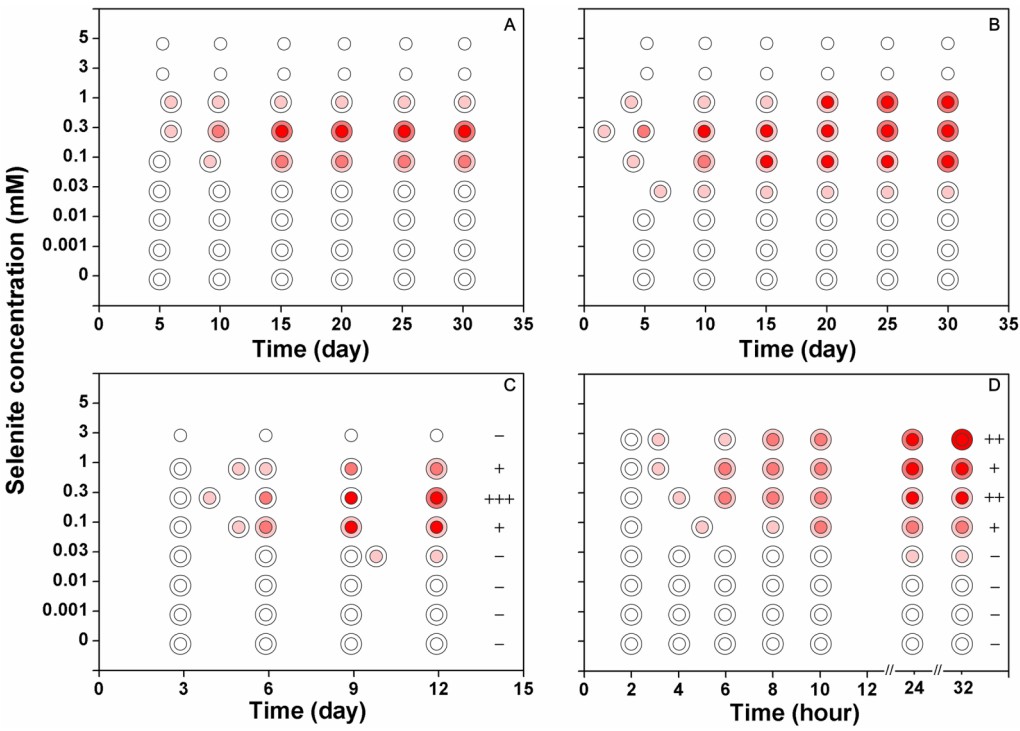

**Figure 3 The reduction intensities of *F. velutipes* supplied with 0–5 mM selenite (indicated by the red colorations of the colonies).** The isolates from the stock colonies were inoculated into the media containing 0–5 mM selenite and were subjected to solid cultivation (A) static cultivation (B) and shaking cultivation (C). In another experiment (D) after shaking cultivation in selenite-free media for 14 days, the resulting mycelial pellets were treated with 0–3 mM selenite. The first pink circle indicates the beginning of the red coloration. The small circle inside represents the central part of the colony (A and B) or the large mycelial pellet (C and D) and the ring outside represents the margin of the colony (A and B) or the small mycelial pellet (C and D). The plus/minus signs on the right side of circles in (C) and (D) represent the intensities of garlic smell after selenite treatment (−: none; +: low; ++: moderate; +++: strong).

diffusion (*Gharieb & Gadd, 2004*). Nevertheless, it was unlikely that the selenite uptake in our study proceeded via adsorption or simple diffusion, since the negatively charged cell membrane would repel $HSeO_3^-$ and the hydrophobic core of lipid bilayer was impermeable to selenite (*Araie et al., 2011*; *Morlon et al., 2006*; *Wells & Richardson, 1985*). Moreover, the desorption procedure adopted in this study following the selenite uptake process further precluded the possibility of abiotic adsorption (see Materials & Methods). Therefore, a carrier-facilitated passive transport may be crucial to the selenite uptake by *F. velutipes*.

Specific selenite transporters have not been identified, and like other trace anions, selenite may share transporters with the major anions such as phosphate (*Lazard et al., 2010*; *Li, McGrath & Zhao, 2008*; *Riedel & Sanders, 1996*), sulfite (*Zhang, Shi & Wang, 2006*) and monocarboxylate (*McDermott, Rosen & Liu, 2010*). We tested the possible roles of phosphate and sulfite transporters in selenite uptake, and assumed that with the up-regulation of phosphate or sulfite transporter after P or S starvation, the rate of selenite uptake would increase (*Li, McGrath & Zhao, 2008*). Opposite to our assumption,

the uptake rate of selenite decreased significantly by ~25% after P and S starvation (Fig. 1C). One explanation is that two transporters with contrasting selectivity for selenite might exist in *F. velutipes*. In *S. cerevisiae*, which is phylogenetically close to *F. velutipes*, the selenite uptake involves the high and low affinity phosphate transporters at low and high (1 mM) phosphate concentrations, respectively (*Lazard et al., 2010*). The high affinity transporter strongly selects phosphate over selenite, whereas the low affinity transporter does not discriminate efficiently between the two anions (*Lazard et al., 2010*). Thus the low affinity transporter can potentially have a higher conductance to selenite than the high affinity transporter (*Lazard et al., 2010*). *Pinson et al. (2004)* observed a close relationship between selenite resistance and the expression of the high affinity phosphate transporter in *S. cerevisiae*. If this transport system applies to *F. velutipes*, the transporter with high phosphate affinity would be up-regulated, whereas the one with low affinity would be down-regulated following P starvation. Meanwhile, the low affinity phosphate transporter would be responsible for the selenite uptake in the control treatment with high phosphate concentration (8.6 mM). Because the high affinity transporter discriminate selenite more efficiently, less selenite is absorbed after P starvation than in the control treatment. Similar sulfite transport system might be responsible for the decrease in selenite uptake after S starvation. Further studies are needed to identify the transporters of selenite in *F. velutipes*.

## Sensitivity of *F. velutipes* to selenite

An overdose of selenite interferes with the S and P metabolism (*Bryant & Laishley, 1988*; *Daniels, 1996*; *Tomei et al., 1995*; *Weissman & Trelease, 1955*) and leads to oxidative stress by reacting with intracellular thiols (*Papp et al., 2007*). In this study, selenite began to be toxic to *F. velutipes* at 0.1 mM (Fig. 2). Based on the colony morphologies, the responses of *F. velutipes* to 0.1 mM selenite were moderate among the commonly cultivated mushrooms (Tables S1, S2 and S6). *Lin et al. (1997)* reported that 0.03 and 0.12 mM selenite started to inhibit the growth of *F. velutipes* in solid and liquid cultivations (PDA media), respectively. The discrepancies with our results in selenite sensitivity may be attributed to the differences in strains and medium components. The corral-like colony margins of *F. veluptipes* when subjected to 1 mM selenite for more than 5 days were also reported when *Fusarium* sp. was exposed to 5 mM selenite (*Gharieb, Wilkinson & Gadd, 1995*). It remains unclear if this is a common response of fungal mycelia when subjected to toxic amount of selenite. The mycelial growth completely stopped at selenite concentrations of 3 and 5 mM (Figs. 2A, 2C and 2D), but the inocula resuscitated without a severe decrease in the growth rate after transferred to selenite-free medium (Fig. 2B; Table S4). Thus, a high concentration of selenite induced the dormancy of the inocula without severely damaging their vitality. Based on our results, we recommend a selenite concentration between 0.03 and 0.1 mM to maintain the balance between Se enrichment and mycelial productivity. This is in agreement with the optimal selenite concentration (0.06 mM) proposed by *Ma et al. (2012)* for the soluble organic Se accumulation during the fermentation of *F. velutipes* mycelia.

## Reduction of selenite to Se(0) by *F. velutipes*

The alteration of colony color and the TEM analysis of the red part of the mycelia confirmed the existence of Se(0) (partly as nanoparticles) when *F. velutipes* was treated with selenite (Figs. 3 and 4). The method of Se(0) determination in this study (based on colony color; *Gharieb, Wilkinson & Gadd, 1995*) was semiquantitative, because the filamentous nature of the fungal mycelia made it difficult for the use of the spectrophotometric method (*Dhanjal & Cameotra, 2010*). Further research is needed to improve the accuracy of Se(0) determination during fungal cultivation. Moreover, the TEM image shows Se(0) particles outside the hyphae of *F. velutipes* (Fig. 4A). It remains unclear whether there was an extracellular pathway for selenite reduction or the Se(0) particles inside the hyphae were leaked out during sample preparation. In addition to yeasts (*Falcone & Nickerson, 1963*; *Gharieb, Wilkinson & Gadd, 1995*; *Nickerson & Falcone, 1963*) and molds (*Brady, Tobin & Gadd, 1996*; *Gharieb, Wilkinson & Gadd, 1995*; *Ramadan et al., 1988*), some species of mushrooms, including *Coriolus versicolor* and *Lentinula edodes*, *Ganoderma lucidum*, *Pleurotus ostreatus* and *Grifola frondosa* (*Gharieb, Wilkinson & Gadd, 1995*; *Vetchinkina et al., 2013*; *Vetchinkina et al., 2016*) have been studied for their abilities to transform selenite to Se(0). In our study, *F. velutipes* and other 11 species of phylogenetically and ecologically varied mushrooms were tested, and 10 of them showed the reducing ability (Fig. S3). Thus it is conceivable that mushrooms commonly possess this reducing ability.

Selenite reduction by fungi to Se(0) has generally been considered as a detoxification mechanism (*Gharieb, Wilkinson & Gadd, 1995*; *Ramadan et al., 1988*; *Vetchinkina et al., 2013*). If the same is true for *F. velutipes*, the reduction intensity would increase as selenite poisoning strengthens. At selenite concentrations below 0.3 mM, the reduction intensities increased with the increasing selenite concentration (Figs. 3A–3C). However, at selenite concentrations of 1 mM or higher, when the mycelial growth was severely inhibited (Fig. 2), the reduction intensities decreased (Figs. 3A–3C). *Lortie et al. (1992)* also reported that selenite poisoning at high concentrations caused a decrease in the reduction rate of selenite by *Pesudomonas stutzeri*. *Nickerson & Falcone (1963)* proposed that the reduction process operated at the expense of the endogenous metabolism. Selenite acts as a prooxidant at high concentrations which undergoes glutathione-mediated reduction to hydrogen selenide (*Izquierdo, Casas & Herrero, 2010*). The depletion of metabolic products, such as glutathione, might be the reason that the Se(0) production decreased at high selenite concentrations, and thus we assume that the reduction of selenite to Se(0) by *F. velutipes* may be metabolism dependent. This idea was tested using mycelial pellets from selenite-free media (14-day-old), as they were supposed to be metabolically active compared with those developed in the media initially containing toxic amounts of selenite. When the mycelia were not constrained by the metabolic activity, the red coloration appeared in 24 h and the reduction intensity increased with the increasing selenite concentrations in the range of 0–3 mM (Fig. 3D). This may explain why the inoculum part of the colony showed faster and denser coloration when subjected to selenite (Figs. 3 and S2). Thus, the reduction of selenite to Se(0) can be influenced by both the selenite concentration and the metabolism activity of the fungi.

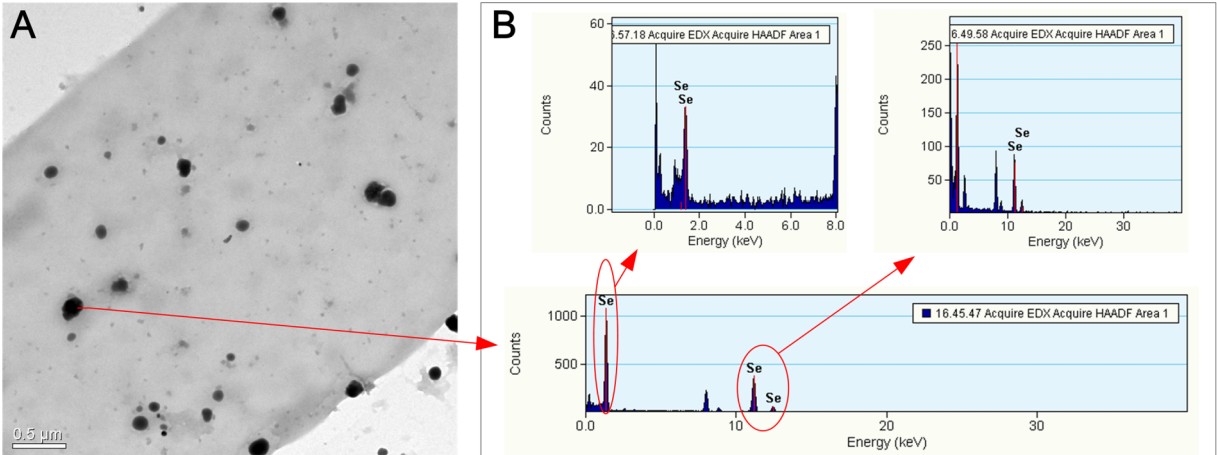

**Figure 4** TEM (A) and EDX (B) images of the hyphae of *F. velutipes* after selenite treatment. The mycelia were picked up from the red region of the 20-day-old colony in the static cultivation supplied with 0.3 mM selenite.

In addition to the red coloration, colonies treated with a toxic amount of selenite often exhibited a garlic smell during cultivation of *F. velutipes* (Figs. 3C and 3D). Similar phenomena have been reported in the literature studying the responses of fungi to selenite, and the garlic smell was suspected to originate from volatile Se-containing compounds, possibly dimethylselenide (*Brady, Tobin & Gadd, 1996*; *Gharieb, Wilkinson & Gadd, 1995*; *Schilling, Johnson & Wilcke, 2011*). Direct evidence is needed to confirm the speciation of the volatile compounds. This may suggest that the reduction of selenite to Se(0) was not the only mechanism for *F. velutipes* to cope with the selenite toxicity. The transformation of selenite to volatile Se species which might appear in this study, together with the reduced uptake (*Gharieb, Wilkinson & Gadd, 1995*) and the reduced incorporation of Se into proteins (*Ramadan et al., 1988*), may also contribute to the tolerance.

## CONCLUSIONS

At near neutral pH, selenite uptake was slightly affected by selenite species and metabolic inhibitor. A carrier-mediated passive transport might be crucial to the selenite absorption by *F. velutipes*. At selenite concentrations of 0.1 mM or higher, the absorbed selenite led to growth inhibition of *F. velutipes*. As a detoxification mechanism, a portion of the selenite was transformed to Se(0) including Se(0) nanoparticles in a metabolic dependent way. These results provided some basic information for the cultivation of the selenized *F. velutipes* and highlighted the opportunity of using mushrooms for the production of Se(0) nanoparticles.

### Funding
This work was supported by the earmarked fund for Modern Agro-Industry Technology Research System CARS-24 and the National Natural Science Foundation of

China (Grant Nos. 41272200 and 41571315). The funders had no role in study design, data collection and analysis, decision to publish, or preparation of the manuscript.

### Grant Disclosures

The following grant information was disclosed by the authors:
Modern Agro-Industry Technology Research System CARS-24: 41272200.
National Natural Science Foundation of China: 41571315.

### Competing Interests

The authors declare that they have no competing interests.

### Author Contributions

- Jipeng Wang conceived and designed the experiments, performed the experiments, analyzed the data, wrote the paper, prepared figures and/or tables.
- Bo Wang conceived and designed the experiments, performed the experiments, contributed reagents/materials/analysis tools, prepared figures and/or tables, reviewed drafts of the paper.
- Dan Zhang performed the experiments, contributed reagents/materials/analysis tools, reviewed drafts of the paper.
- Yanhong Wu analyzed the data, reviewed drafts of the paper.

### Data Deposition

   The raw data has been supplied as a Supplemental Dataset Files.

### Supplemental Information

Supplemental information for this article can be found online at http://dx.doi.org/10.7717/peerj.1993#supplemental-information.

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
