# Peer review of "Selenium uptake, tolerance and reduction in Flammulina velutipes supplied with selenite"

_PeerJ, doi:10.7717/peerj.1993_

## Round 0.1 · original submission · Minor Revisions

As you can see from the reviews, the manuscript needs minor revisions. Kindly submit your articles as per the suggestions made by the reviewers.

Reviewer 1 ·

Basic reporting

No comments

Experimental design

No comments

Validity of the findings

No comments

Additional comments

After reading the whole manuscript, the main fermentation conditions of Flammulina velutipes were investigated. Thus, the "toxicity" was not appropriate, please re-writted the Title.

Reviewer 2 ·

Basic reporting

In my opinion the manuscript is well-prepared:
-the English language is professional
-the structure of the manuscript is correct
-the figures are well labeled and their quality is good
- the literature is well referenced (missing only a few recent literature points concerning the research area)
-in my opinion the manuscript confirms PeerJ standards

Experimental design

In appreciate undertaken research topic within the scope of the Journal.
The research question is well defined – the problem of the transport of selenites to the mushroom cell and its metabolism is not well understood.
Disadvantage of the manuscript is imprecise description of the methods of research.
For example, the method of determining of selenium reduced to 0 oxidation is imprecise and poorly described.
In my opinion the quantification would be more accurate. There are simple spectrophotometric methods for such determination, known for decades. The results presented in the manuscript the Authors should present as preliminary, declaring further research on this problem. I hope that the authors do not intend to end the studies at the preliminary results (?)

Validity of the findings

As sated above, described in the manuscript research represents the development of extensively examined in the recent years problem of metabolism of selenium in higher fungi. The main goal of this experiments is to obtain a new kind of food supplement for use in cancer prevention. The preliminary results (in-part cited in the “Introduction” of the revised manuscript) show, that the studies are justified from a “medical point of view”.
The results of the research presented in the manuscript constitute a contribution in these studies.

Additional comments

Described in the manuscript research focuses on interesting problem. The manuscript may be published after minor modifications, regarding the clarification of the methodology (determination of Se (0), and analysis in the "Discussion" accuracy (precision) of results, in my opinion in-part preliminary.

Reviewer 3 ·

Basic reporting

Authors have investigated the uptake, toxicity and reduction of selenite in
a widely cultivated mushroom, Flammulina velutipes. The work carried out by the authors is interesting and can be used in the production of Se(0) nanoparticles.

Experimental design

The experimental design is appropriate and authors has attempted required methodology for the proposed work

Validity of the findings

No

Additional comments

The findings presented in the present manuscript, however i would suggest authors to synthesize the nanoparticales and see the toxicity to validate the reports.

·

Basic reporting

This study is interesting, and research findings are of valuable to those in the selenium research field. The manuscript was generally well written, including required information.

Experimental design

The research questions were clear, and the experimental design was generally scientifically sound.

Validity of the findings

The research conclusions were generally supported by the results, except for the statement regarding elemental selenium nanoparticles (see my comments below).

Additional comments

Specific Comments:
l. 24-25: The statement is not valid - using mushrooms to produce elemental selenium nanoparticles. In other words nanoscale selenium particles can be produced by F. velutipes, but not all biogenic elemental selenium particles are in nanoscale.
l. 44-46: awkward sentence; revision needed.
l. 50-51: "are safer and more bioavailable" to what organisms? Are the two cited references relevant?
l. 67: "used by macronutrients" - any macronutrients or specific nutrient elements?
l. 82: Change it to due to the formation of elemental Se (0); elemental selenium includes both bulk elemental selenium and nanoscale elemental selenium. This can be seen in Figure 4.
l. 114: "uptake solution"? Did you mean "cultural solution"?
l. 150-151: "solid medium" and "liquid medium"? Change it to (plate culture) and (liquid cultural)?
l. 168: How to determine "the full development" status?
l. 203-204: Not clear. Revision needed
l. 291-291: repeated sentence (see lines 199-200)
l. 303-304: Again, the red color can also be due to the formation or presence of bulk elemental selenium in the solution. In Figure 4A there are many large selenium particles of approximately 200-300 nm.
l. 209-311: what are the "other 11 species"? They were not mentioned in the method section. Where is Figure S3?
l. 337: There was no "volatile Se species observed in this study". "A garlic smell" can not be overstated as "observed".
l. 344-347: Change the statement to "selenite can be reduced to elemental selenium including nanoscale selenium particles".
l. 347: See my comments above. Revision needed.

Reviewer 5 ·

Basic reporting

Please see attached file.

The authors highlighted the major findings of Se supplementation to mushroom in a very straightforward manner.

Experimental design

Please see attached file.

The experimental design is described thoroughly, and includes descriptions for the process of incubating the mushroom with different Se(IV) concentrations and at different pH conditions.

Validity of the findings

Please see attached file.

Annotated reviews are not available for download in order to protect the identity of reviewers who chose to remain anonymous.

---

## Round 0.2 · accepted · Accept

Revised Mss. is up to the standard of the journal. I recommend publication in PeerJ.